# Biallelic and gene-wide genomic substitution for endogenous intron and retroelement mutagenesis in human cells

Tomoyuki Ohno[1,2], Taichi Akase[1], Shunya Kono[1], Hikaru Kurasawa[1,3], Takuto Takashima[1], Shinya Kaneko[1] & Yasunori Aizawa ![ORCID] [1,2,3✉]

Functional annotation of the vast noncoding landscape of the diploid human genome still remains a major challenge of genomic research. An efficient, scarless, biallelic, and gene-wide mutagenesis approach is needed for direct investigation of the functional significance of endogenous long introns in gene regulation. Here we establish a genome substitution platform, the Universal Knock-in System or UKiS, that meets these requirements. For proof of concept, we first used UKiS on the longest intron of *TP53* in the pseudo-diploid cell line HCT116. Complete deletion of the intron, its substitution with mouse and zebrafish syntenic introns, and specific removal of retrotransposon-derived elements (retroelements) were all efficiently and accurately achieved in both alleles, revealing a suppressive role of intronic *Alu* elements in *TP53* expression. We also used UKiS for *TP53* intron deletion in human induced pluripotent stem cells without losing their stemness. Furthermore, UKiS enabled biallelic removal of all introns from three human gene loci of ~100 kb and longer to demonstrate that intron requirements for transcriptional activities vary among genes. UKiS is a standard platform with which to pursue the design of noncoding regions for genome writing in human cells.

[1] School of Life Science and Technology, Tokyo Institute of Technology, Yokohama 226-8501, Japan. [2] Logomix, Inc., Tokyo 104-0053, Japan. [3] Kanagawa Institute of Industrial Science and Technology, Ebina 243-0435, Japan. ✉email: yaizawa@bio.titech.ac.jp

During evolution, noncoding sequences have accumulated within mammalian genomes, where they now represent >90% of the sequences[1,2]. As compared with the genomes of bacteria and lower eukaryotes, mammalian genes are separated by larger intergenic sequences and also contain many more and longer introns. As part of the Encyclopedia of DNA Elements (ENCODE) project, chromatin-immunoprecipitation followed by sequencing (ChIP-seq) and its modified methods showed the association of histone marking and/or transcriptional factor binding to ~80% of the human genome[3]. However, the importance of these biochemical activities for the regulation of a particular gene or cell function remains largely untested in the context of the native chromatin architecture, due to the absence of a systematic mutagenesis method for the vast noncoding regions[4,5].

A special approach is required for mutagenesis of human introns, which are 3.4 kb in length on average[1] and cover ~25% of the human genome[6]. With the development of clustered regularly interspaced short palindromic repeat (CRISPR)/Cas systems, the in vitro mutagenesis of reporter plasmids containing only the target intron and its flanking exons has been replaced by direct mutagenesis of the endogenous intron in living cells[7,8]. Point mutations and/or small insertion/deletions (indels) can be site-specifically introduced at the endogenous intronic loci to allow the investigation of their effects in living human cells. However, the CRISPR/Cas9 system does not allow for the targeting of any and all nucleotide positions, as it requires a protospacer adjacent motif (PAM)[9]. Thus, the accurate removal of only intronic regions by a pair of guide RNAs (gRNAs) targeting both 5' and 3' intron-exon junctions is practically impossible because of limited PAM repertoires. Moreover, if multiple different gRNAs are used to mutate several positions in the target introns, more off-target effects should be expected[10].

Retroelements pose an even greater challenge for mutagenesis by CRISPR. Retroelements have very similar sequences and are present throughout the human genome, especially within introns, where they are heavily accumulated. Their sequence similarity prevents locus-specific mutagenesis with CRISPR. Thus, another strategy is typically used, homologous recombination (HR)-based replacement, in which target sequences containing mutations at any desired locus can be incorporated into the genome. This approach must, however, be carried out in a scarless manner for studies on intron functionality, as there are no regions within and around introns that are safe from the effects of scar sequences, such as drug selection markers and loxP-type elements, on the expression of the target gene and its surrounding genes[6,11].

Importantly for autosomal genes in diploid genomes, the introduction of identical mutations in both alleles is preferable for characterizing the influence of mutations on gene expression and the downstream phenotypes. In the present study, we describe a genome engineering platform, called Universal Knock-in System or UKiS, that fulfills all of these requirements for comprehensive and systematic mutagenesis of vast noncoding genomic regions in human cells. UKiS is based on homology-dependent genome modification with the two-step positive and negative selection process. The feasibility and applicability of UKiS were first tested in a pseudo-diploid cancer cell line, HCT116, by targeting the longest intron in TP53 (the first intron, 10.8 kb), which has been heavily targeted by retrotransposons during the evolution of the mammalian genome. UKiS was used to efficiently incorporate a variety of mutations in the intron: deletion of the entire intron, its substitution with the corresponding mouse or zebrafish intron, and precise removal of all or specific retroelement types within the intron. Characterization of these mutant cells revealed roles of intronic and retroelement sequences in TP53 expression. UKiS was also used with human induced pluripotent stem (iPS) cells and for removal of all introns from three human genes of ~100 kb or more in length. This study demonstrates that UKiS is advantageous for the direct investigation of functional impacts of long endogenous noncoding regions in diploid genomes.

## Results

**Design of UKiS.** In general, mutagenesis for long endogenous regions can be performed by a one-step knock-in approach via HR. Thus, a selection process is required to obtain cell clones with the desired mutations efficiently. In UKiS, we set up a two-step HR/selection procedure (Fig. 1a). In the first step, both target alleles are replaced with either one of two different custom-made UKiS donors (Fig. 1b). Both UKiS donors have homology arms at both ends and, between the arms, a positive and negative dual selection marker is introduced. This marker expresses a chimeric protein consisting of green fluorescent protein (GFP) and a drug resistance gene product specific for puromycin or blasticidin, each of which is linked to the GFP by a T2A self-cleaving peptide sequence. The marker gene is flanked by insulator sequences that promote its stable expression. The drug resistance genes are used for positive selection with both puromycin and blasticidin during the first step of UKiS, so as to obtain cells containing one of each UKiS donor substituted for the target region within the two alleles. To facilitate the HR reaction, a double-stranded break is induced by CRISPR at the chromosomal position that partially overlaps the region targeted by the homology arm.

In the second step of UKiS, a DNA payload plasmid containing the mutated target sequence ("mutating payload") replaces the UKiS donors in both alleles. The replacement is promoted by CRISPR cleavage with an Off-Target Less gRNA (TL-gRNA) at the site adjacent to the right arm on the UKiS donor. The TL-gRNA sequence (5'- GGCGCAACGCGATCGCGTAA) has no significant similarity with any other sequences in the human genome to minimize the off-target risk[12]. Cells with successful replacement in both alleles lose GFP expression and can thus be collected by fluorescence-activated cell sorting (FACS).

**Replacement of *TP53* first intron alleles with UKiS donors.** To evaluate the feasibility and advantages of UKiS, we first used it to mutagenize the first intron of TP53. TP53 is one of the most studied human genes: it safeguards against tumor initiation and progression in many organs[13–15]. The longest variant of the TP53 transcriptional unit is 19.2 kb and has 11 exons, with the first intron being 10.8 kb in size (Fig. 2a). Among all the introns of human genes, this intron ranks in the top 12 percentile in terms of length (Supplementary Fig. 1). Typically, the first intron in many human genes is more highly enriched for gene regulatory elements relative to other introns[16,17]. Likewise, significant histone modifications for transcriptional activation including H3K27 acetylation and H3K4 trimethylation and binding sites for various transcription factors were observed in the first intron of TP53, as shown by the ENCODE project[3]. In addition, this intron largely comprises retroelements, with 21 independent insertions of non-long-terminal-repeat-type retrotransposons (long and short interspersed nuclear elements, LINEs and SINEs), which account for 8.0 kb in total and occupy 74% of this intron. For our TP53 mutagenesis, we first used a human colon cancer cell line, HCT116, which is pseudo-diploid and thus has two TP53 loci[18]. The left arm of both UKiS donors is 809 bp long and corresponds to sequences upstream of exon 1 (Fig. 2a). In contrast, the genomic region targeted by the right arm spans the second and fourth exons of TP53 and partially overlaps the sequence targeted by the TP53-specific gRNA, gRNA-TP53(R), that was used in the first step of UKiS as described below (Fig. 2a).

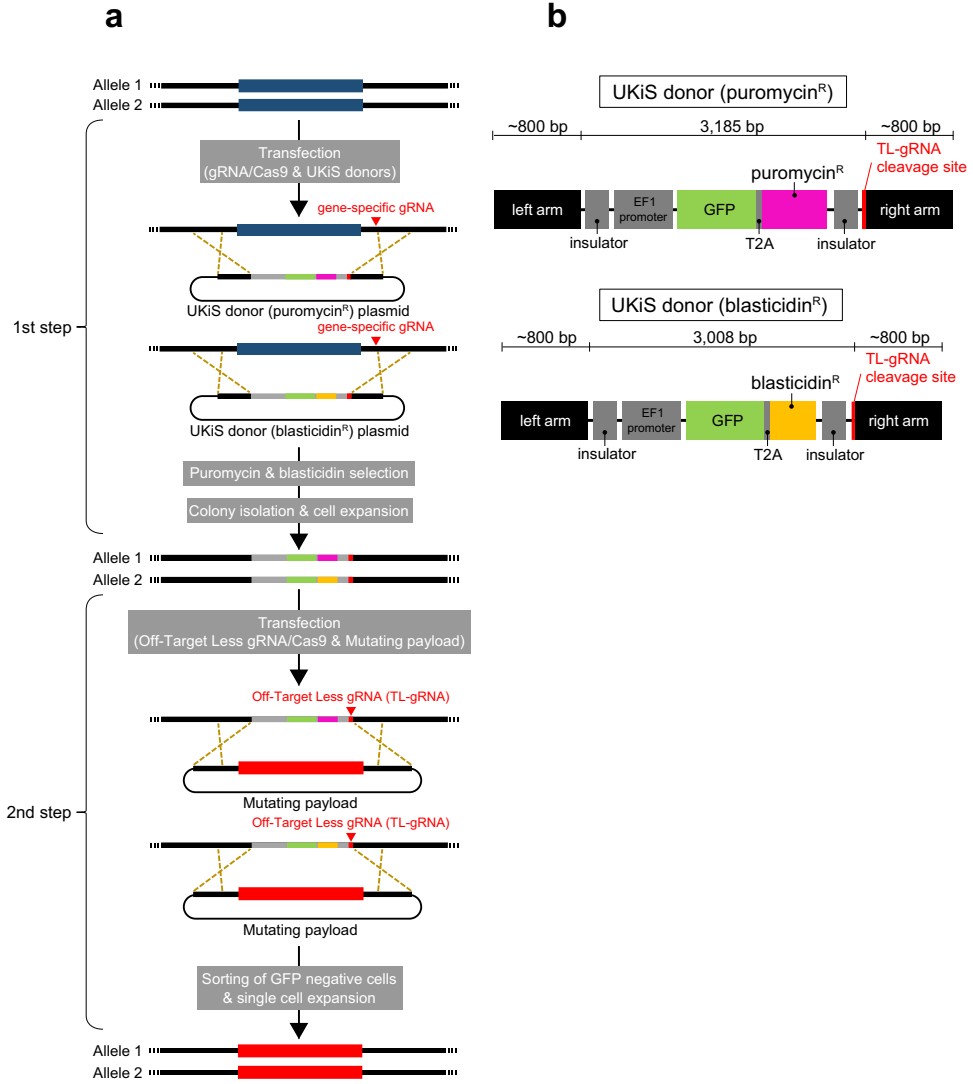

**Fig. 1 Overview of UKiS. a** The two-step process of UKiS. First step: co-transfection of the expression plasmid for Cas9 protein and the gene-specific gRNA and the two UKiS donor plasmids, followed by puromycin and blasticidin selection, to collect cells that had undergone biallelic replacement of the target locus with UKiS marker sequences. Second step: co-transfection of the expression plasmid for Cas9 and the Off-Target Less gRNA (TL-gRNA) and the mutating payload plasmid containing the desired mutation, followed by fluorescence-activated cell sorting (FACS) to collect cells that do not express GFP because UKiS marker sequences at both target alleles were replaced by the payload sequence. **b** Schematic illustration of the two UKiS donors. The UKiS donors consist of homology arms and insulators on both sides and, between them, a marker gene encoding a chimeric protein consisting of GFP and one of two antibiotic markers (puromycin or blasticidin), each of which is linked by a T2A self-cleaving peptide sequence. The sequence corresponding to TL-gRNA[12] partially overlaps with the right arm of the donor sequence.

For the first step of UKiS, the two UKiS donors and the gRNA/Cas9 expression vector were co-transfected into HCT116 cells, followed by dual selection using puromycin and blasticidin (Fig. 2b). Of the 15 clones obtained after the drug selection, 8 clones were confirmed by junction genotyping PCR to have lost the first intron and instead to have both a puromycin and blasticidin UKiS donor individually inserted in each of the two alleles (Fig. 2c). As a control, in a separate experiment, only the puromycin UKiS donor was co-transfected with the gRNA/Cas9 expression vector, leading to unsuccessful biallelic replacement in the 10 surviving clones after puromycin selection (Supplementary Fig. 2). Moreover, we experimentally verified that no off-target cleavage occurred in the eight obtained clones. The genomic regions on chromosomes 2 and 16 that contain the sequences most similar to the targeting sequence of the gRNA used here had no insertions or deletions and thus were unlikely to be cleaved by this gRNA (Supplementary Fig. 3).

**Replacement of the UKiS donor alleles with the wild-type and mutant introns**. In subsequent experiments related to *TP53* mutagenesis in HCT116 cells, one of the clones obtained here after the first step (clone #1–10) was generally used to create the various mutants in the second step of UKiS. Flow cytometry confirmed that almost all the cells (99.7%) of clone #1–10 were positive for GFP fluorescence as expected (Fig. 3a). The UKiS donor-integrated alleles in clone #1–10 were first replaced with a payload containing the wild-type first intron of human *TP53* by proceeding to the second step of UKiS (Fig. 3b). The GFP-negative cells were obtained by FACS at a significant efficiency (2.24%), followed by cloning and PCR genotyping with primer pairs covering the entire integrated region. Seven of the eight cell clones resulted in PCR products of the expected length, 14.0 kb. To assess any off-target effects of the TL-gRNA, the locus most similar to the target sequence of this gRNA, which is located on chromosome 19, was sequenced in all seven hit clones, confirming that

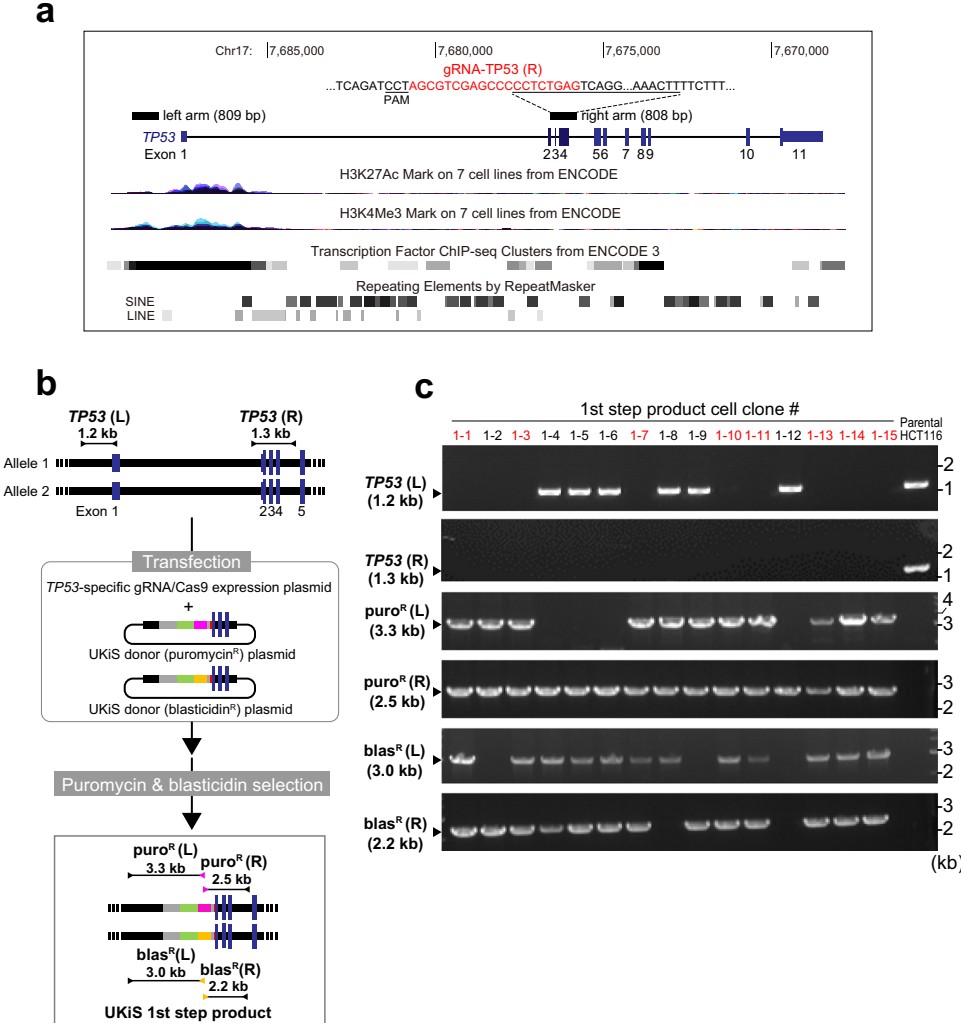

**Fig. 2 The first step of UKiS: replacement of the first intron of *TP53* with the UKiS donor. a** Graphical representation of the human *TP53* locus from the UCSC genome browser[41], indicating one *TP53* mRNA variant (NM_000546) and the histone activation mark (H3K27Ac and H3K4Me3), and transcription factor binding regions, all of which were identified by the ENCODE project[3], and retrotransposon-derived elements (LINEs and SINEs are long and short interspersed nuclear elements, respectively). The regions targeted by the homologous arms of the UKiS donors are denoted by black boxes. The same arms were also used for the mutating payloads during the second step of UKiS. The gRNA-TP53(R) target sequence is highlighted in red and the PAM sequence part is underlined within the sequence of the right arm part. **b** Schematic diagram of the first step of UKiS for the first intron of *TP53*. Both UKiS donor plasmids, the marker parts of which are indicated by the same colors as shown in Fig. 1b, were transfected into HCT116 cells, leading to isolation of cell clones that had undergone homologous recombination within the *TP53* locus after dual selection with puromycin and blasticidin. Horizontal lines flanked by two arrowheads represent the target regions for the junction genotyping PCR. **c** Representative gel image of the junction genotyping PCR to confirm deletion of the *TP53* first intron and insertion of the UKiS marker, with the expected length of PCR genotyping amplicons indicated. Of the 15 selected clones, 8 had successful replacement of the intron with the UKiS donor in both alleles and are highlighted in red. Source data are provided as a Source Data file.

this off-target candidate site was not mutated (Supplementary Fig. 4). We then used the same strategy to acquire six clones with deletion of the entire first intron (Fig. 3c), four clones in which the first intron was replaced with the mouse *Tp53* first intron (Fig. 3d), and four clones in which it was replaced with the zebrafish *tp53* first intron (Fig. 3e), as indicated by PCR amplification of the target regions that resulted in the expected lengths.

To confirm biallelic modification in the obtained clones, we used the presence of heterozygous single-nucleotide polymorphism (SNP) sites. Our direct sequencing of genomic PCR amplicons was first performed to identify any heterozygous SNPs in the *TP53* locus. We found a known SNP at rs12947788 that is heterozygous (C or T) in the parental HCT116 cells (Fig. 4a). Next, sequencing of the genomic PCR amplicon that spans from the UKiS donor region to a position downstream of rs12947788

allowed us to confirm that the C and T alleles at rs12947788 in clone #1–10 had the puromycin and blasticidin versions of the UKiS donor, respectively (Fig. 4b). Finally, the biallelic replacement with the human, mouse, zebrafish, and null introns in three individual HCT116 clones each was also confirmed by the SNP genotyping at rs12947788 on PCR amplicons including the entire replaced regions between both arms (Fig. 4c and Supplementary Fig. 5). Thus, UKiS efficiently accomplished the biallelic mutagenesis of the 10.8 kb endogenous intron.

**Evaluating the influence of the entire deletion or syntenic substitutions of the first intron of *TP53* on its expression.** The HCT116 clones with successful biallelic substitution were next characterized to investigate the influence of these modifications on

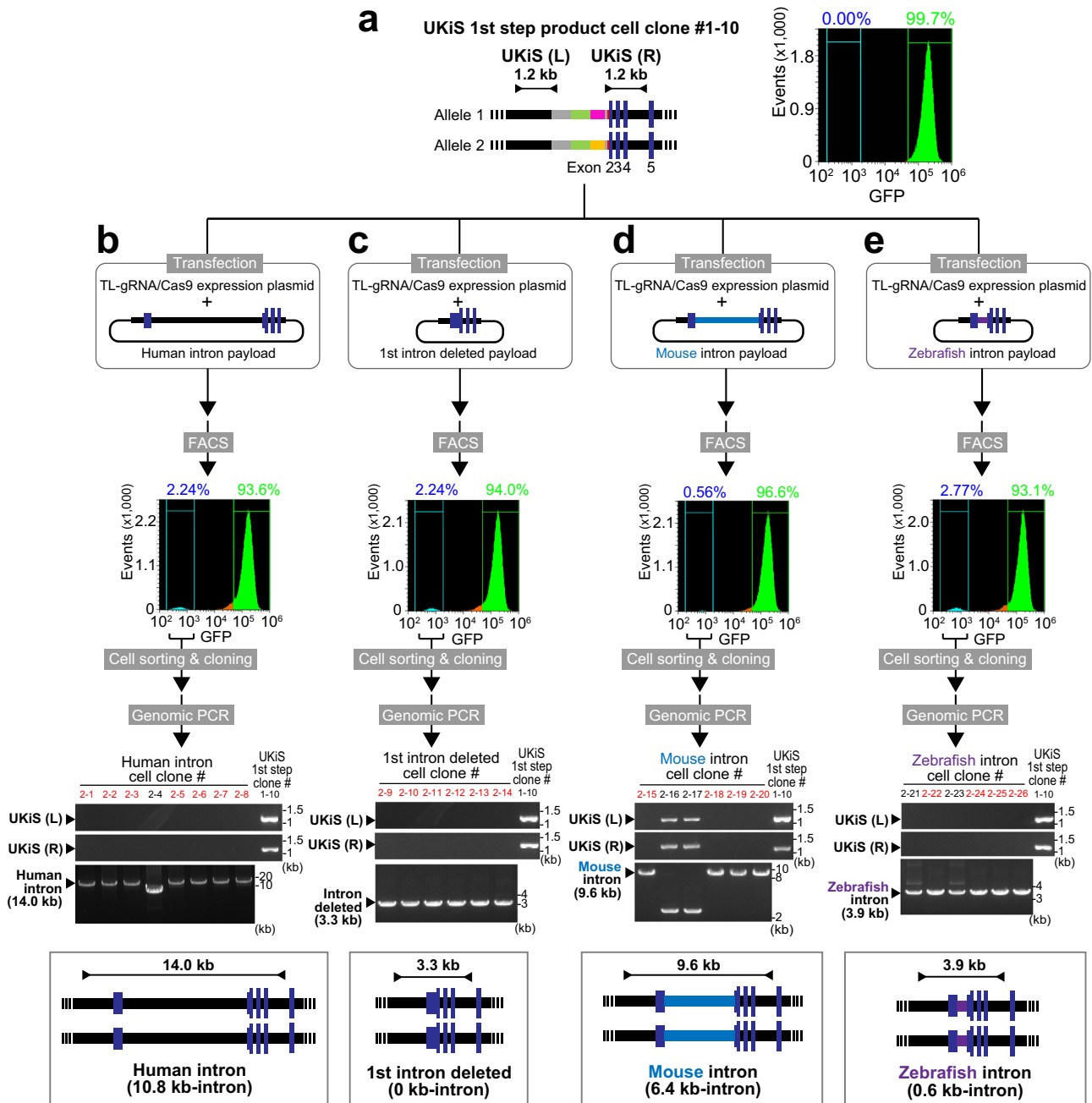

**Fig. 3 The second step of UKiS: replacement of UKiS donor alleles in the first intron of *TP53* with either the wild-type first intron or one of three mutant first introns. a** Schematic diagram of UKiS donor alleles in clone #10 (#1–10), the marker parts of which are indicated by the same colors as shown in Fig. 1b. Horizontal lines flanked by two arrowheads represent the target regions for the junction genotyping PCR performed below. Flow cytometric analysis of GFP fluorescence in clone #1–10. Replacement of both UKiS donor alleles with synthetic introns (**b**: human *TP53* full-length intron, **c** the entirely deleted intron, (**d**) syntenic mouse intron, (**e**) syntenic zebrafish intron) in clone #1–10. First, the mutating payload plasmid and TL-gRNA/Cas9 expression plasmid were transfected into clone #1–10. Thereafter, GFP-negative cells were collected by FACS and cloned. Biallelic substitution of UKiS markers with the mutating payload plasmid was confirmed by junction genotyping PCR that targeted the regions represented by horizontal lines flanked by two arrowheads in the schematic diagrams of the *TP53* locus after successful replacement, with the expected length of PCR genotyping amplicons indicated. In the agarose gel images, lane numbers of clones that underwent successful recombination in both alleles are in red. Source data are provided as a Source Data file.

*TP53* transcription in the context of the native chromatin architecture. RT-PCR using primers for the first and fourth exons was performed to demonstrate that the human, mouse, and null introns gave rise to a single PCR band of a similar size in every case, whereas the zebrafish intron resulted in two bands (Fig. 5a). DNA sequencing of all the RT-PCR products indicated that the single bands from the human and mouse introns and the slower-migrating main band from the zebrafish intron corresponded to the splicing products lacking the original first intron. Given the overall poor sequence homology of the syntenic introns of the three vertebrates, this high convertibility suggests that these introns may share short *cis*-element(s) for proper splice site determination. Sequencing of the minor band from the zebrafish intron indicated that a different 5′-splice site had been used within the first exon

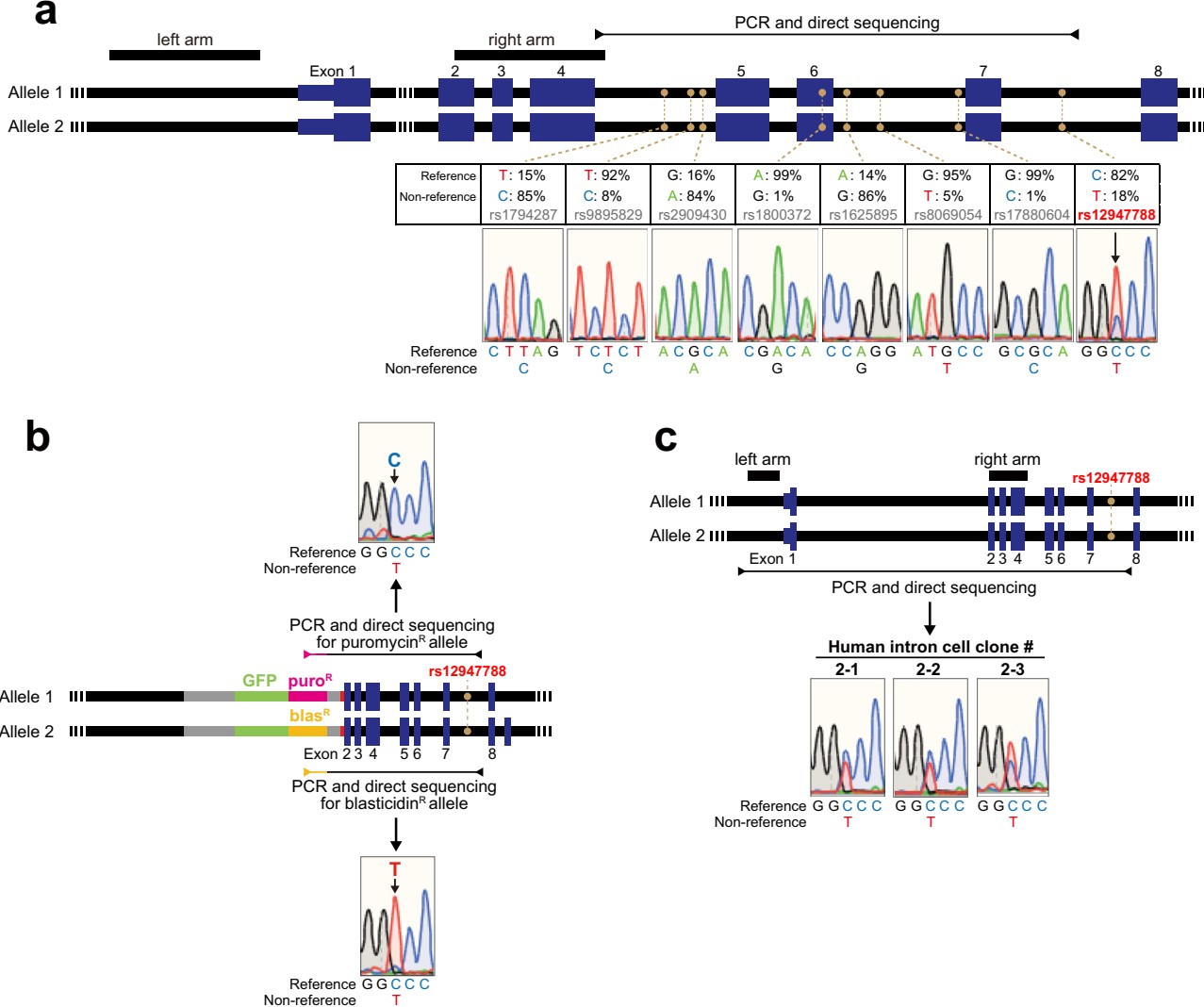

**Fig. 4 Validation of biallelic replacement of the first intron of *TP53* in the first and second step of UKiS. a** Graphical representation of the human *TP53* locus, indicating the positions and allele frequencies of eight common SNPs (filled orange circles; from dbSNP151[42]). Among them, direct sequencing of PCR amplicons from genomic DNA of the parental HCT116 cells demonstrated that the SNP site rs12947788 is heterozygous in HCT116 (black arrow). Black boxes represent homology arms used in our UKiS mutagenesis to *TP53*, and the horizontal line flanked by two arrowheads represents the target region for PCR and subsequent sequencing. **b** Genotyping of clone #1–10, which was used to create all the *TP53* mutant clones in this study. Allele-specific PCR was performed by using primers for puromycin or blasticidin marker sequences. The puromycin and blasticidin alleles had C and T at rs12947788, respectively. Horizontal lines flanked by two arrowheads represent the target region for the PCR of each allele. The heterozygous SNP site, rs12947788, is indicated within the 7th intron with filled orange circles. **c** For human wild-type intron clones #2–1, #2–2, and #2–3, graphical representation of the human *TP53* locus is shown on the top: black boxes represent the positions of homology arms used in our UKiS mutagenesis to *TP53*, the horizontal line flanked by two arrowheads represents the target region for PCR, and the filled orange circles denote the heterozygous SNP site rs12947788. Direct sequencing of the PCR genotyping amplicons indicated double peaks only at rs12947788 on the resultant sequencing chromatograms for these three clones.

(Fig. 5b). As this alternate 5′-splice site has the consensus sequence (AG/GT)[11], it is likely that the human and mouse introns, but not the zebrafish intron, have in common a functional element(s) that prevents the use of this exonic position as the 5′-splice site.

In addition to the qualitative analyses on splicing fidelity, the mutant cell clones were also used for quantitative characterization of the mutation effects on *TP53* transcription and translation. As compared with the human intron, real-time RT-PCR showed that deletion of the entire intron elevated the *TP53* transcription by at least three-fold (Fig. 5c), which also led to an increase in *TP53* translation by 1.9-fold as shown by immunoblotting (Fig. 5d, e). These data clearly indicate that the histone modifications and binding of various transcription factors that have been previously detected within this intron[3] are not necessary for the basic

transcriptional activity of *TP53*. To a similar extent, human-to-mouse intron swapping elevated the transcription and translation of *TP53* (Fig. 5c, e). We thus speculate that certain element(s) present only in the human intronic sequence may play a suppressive role in *TP53* transcription.

**Removal of retroelement- and *Alu*-derived sequences from the first intron of *TP53*.** To identify suppressive *cis*-elements in the first intron of *TP53*, we created additional intron mutants with a focus on retroelements. According to RepeatMasker[19], retroelements are responsible for most sequence differences between the human and mouse first introns, especially the primate-specific SINE *Alu* elements, which occupy 47% of the first intron of *TP53*. As retrotransposons that have jumped into intergenic regions of

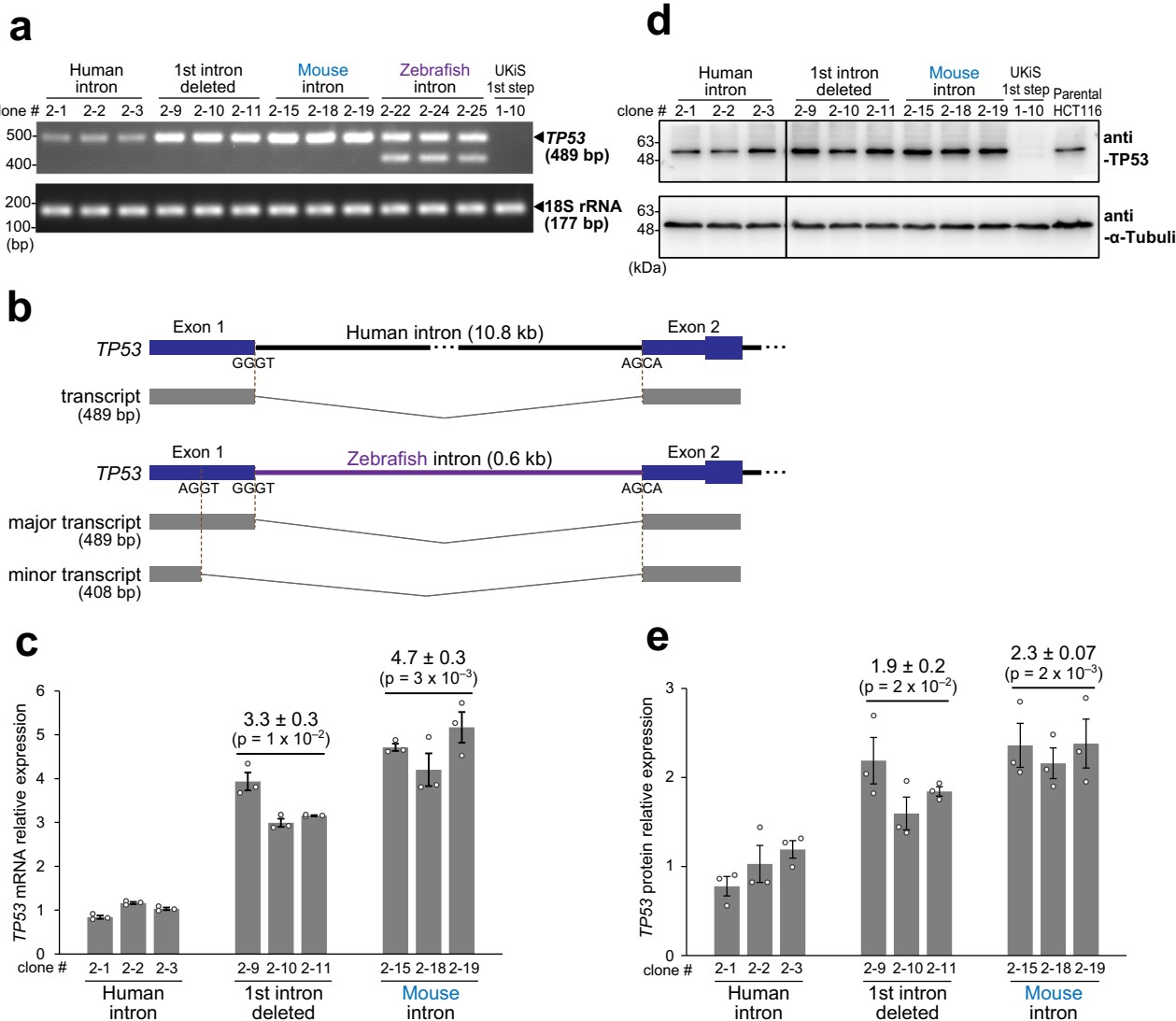

**Fig. 5 Evaluating the influence of intron deletion and syntenic substitution of the first intron of *TP53* on its expression. a** Representative image of agarose gel electrophoresis for RT-PCR products targeting the *TP53* mRNA from three clones of the four variants of the first intron of *TP53*: human wild-type intron, intron deletion, mouse intron, and zebrafish intron (clone numbering in this figure is as in Figs. 3 and 4). **b** Schematic drawing of splice site selection of the *TP53* first intron in the cell clones with the wild-type human intron (top) and the zebrafish intron (bottom). In the HCT116 cells, the zebrafish intron leads to use of the same 5′ splice site as the human wild-type intron does in most cases, but an AGGT site within the first exon is also used as a minor 5′ splice site. **c** Real-time RT-PCR of *TP53* mRNA. 18S rRNA was used as an internal control. Reactions were run in duplicate in three independent experiments. Clones with the entirely deleted intron and syntenic mouse intron showed increased *TP53* mRNA expression as compared with those with the human wild-type intron. The relative fold-changes were calculated by the ΔΔCt method. **d** Representative image of immunoblotting for *TP53* translational expression in the clones shown in (**a**). Blots of TP53 and α-Tubulin (internal control) were derived from samples run on parallel gels. The blot images for TP53 and α-Tubulin were cropped from each of the membrane (some regions were removed for clarity). Blotting was performed in three independent experiments. **e** Quantification of *TP53* translational expression from the immunoblotting experiments shown in (**d**). Bands were quantified by ImageJ[43]. Normalization was performed using α-Tubulin expression. Data in (**c**) and (**e**) represent the mean ± SD of three independent experiments for the individual clones, and the *p*-values (in parentheses; relative to the human intron clones) were calculated by a two-tailed Student's t-test. Reproducibility of (**a**, **c**, **d**, and **e**) was confirmed in three independent experiments. Source data in (**a**, **c**, **d**, and **e**) are provided as a Source Data file.

vertebrate genomes can become distal *cis*-elements for the expression of downstream genes[20,21], it was plausible that some of these intronic *Alu* elements may function as internal *cis*-elements for *TP53* expression. Clone #1–10 was again used for the second step of UKiS to create HCT116 clones with a retroelement-free and *Alu*-free first intron of *TP53* (Fig. 6a).

With the human intronic sequence as a starting point, our in silico design removed the base pairs that RepeatMasker indicated had originated from retrotransposons or *Alu* elements and then connected the remaining base pairs seamlessly. The designed

retroelement-free and *Alu*-free introns were synthesized by overlapping PCR from 7 and 12 PCR amplicons of genomic DNA, respectively (Fig. 6a). The assembled introns were precisely introduced between the first and second exons in the mutating payload plasmids by Gibson Assembly[22]. After transfection of these plasmids into clone #1–10, the GFP-negative cells were sorted and cloned, allowing us to obtain mutants with either a retroelement-free first intron or an *Alu*-free first intron in an efficient and biallelic manner (Supplementary Fig. 6), which was confirmed again by SNP analysis at rs12947788 (Supplementary

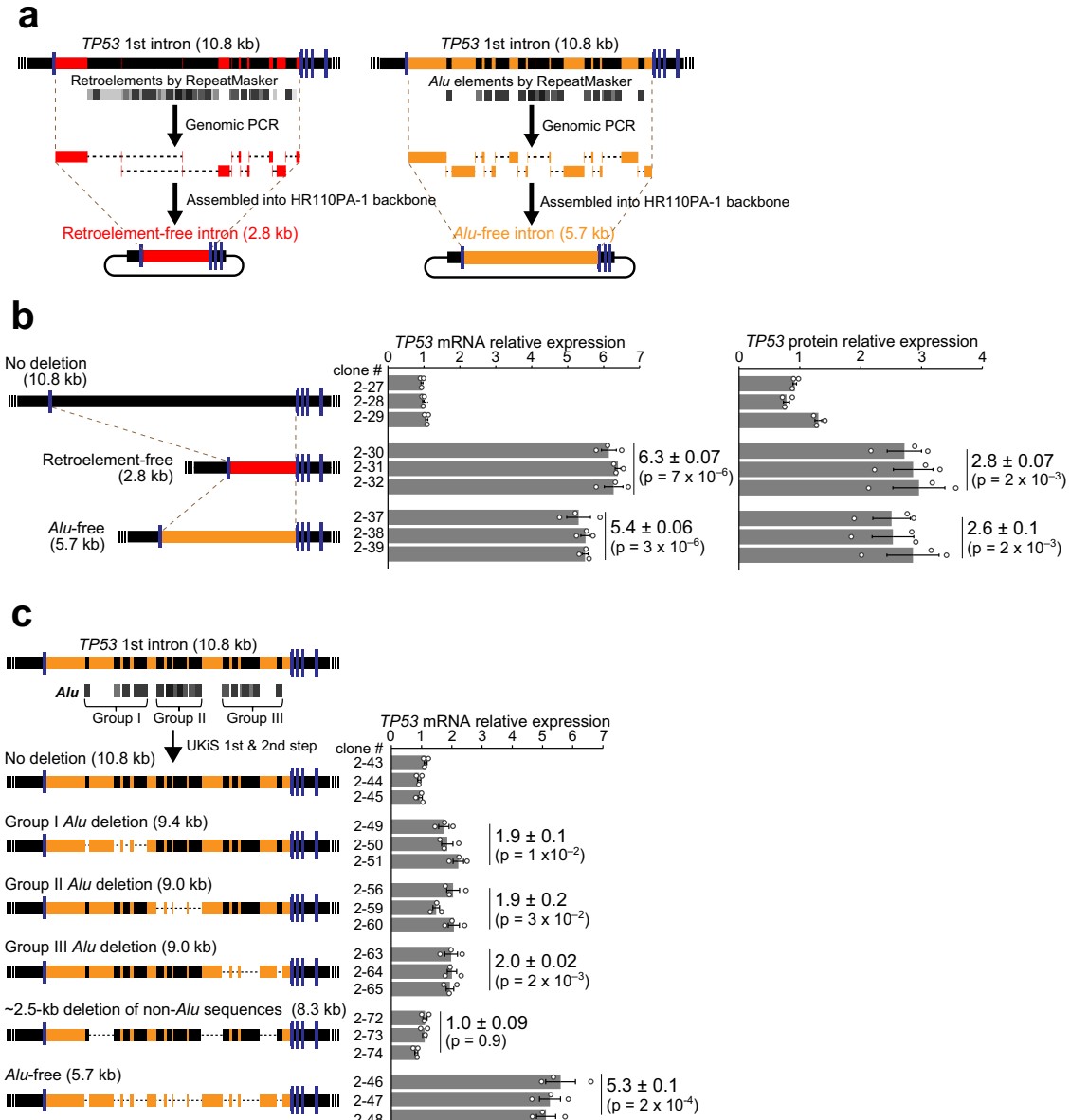

**Fig. 6 Removal of retroelement and *Alu* sequences from the endogenous first intron of *TP53*. a** Graphic representation of the synthesis of retroelement- and *Alu*-free versions of the *TP53* first intron. Non-retroelement and non-*Alu* sequences are shown in red and orange, respectively. **b** Transcriptional and translational impacts of retroelement and *Alu* removal on *TP53* expression. **c** Transcriptional impacts of partial removal of *Alu* elements and removal of non-*Alu* sequences on *TP53* transcription. The removed regions are denoted by dashed lines in the schematic drawings of the 5 deletion mutants. In (**b**) and (**c**), three clones of each mutant cell were subjected to real-time RT-PCR (**b**, **c**) and immunoblotting (**b**). 18S rRNA was used as an internal control for real-time RT-PCR. Relative fold-changes were calculated by the ΔΔCt method. For immunoblotting, α-Tubulin was used as an internal control. Data in (**b**, **c**) represent the mean ± SD of three independent RT-PCR experiments, and the *p*-values (in parentheses; relative to the human intron clones) were determined by a two-tailed Student's t-test, based on three independent experiments for the individual clones. Source data in (**b**, **c**) are provided as a Source Data file.

Fig. 7). As speculated, transcription and translation of *TP53* were significantly elevated in both types of mutants (Fig. 6b), suggesting that intronic *Alu* sites act as a suppressive *cis*-element in this context. The increased level of *TP53* expression after deletion of the *Alu* clusters is sufficient to make a substantial impact on the transcriptional expression of *TP53* downstream genes that regulate growth arrest (*CDKN1A*), apoptosis (*BAX* and *PUMA*), and DNA repair (*RRM2B* and *GADD45A*)[23] (Supplementary Fig. 8), suggesting that the intronic *Alu* sites have a substantial impact on signaling in the *TP53* pathway via tuning *TP53* expression.

As at least 17 independent *Alu* insertions are recognized in this intron according to RepeatMasker, one can speculate that the

suppressive effect may be due to any one of them. To assess this possibility, we divided the *Alu* elements into three groups based on their locations within the intron and then deleted the *Alu* elements in one group at a time by following the second step of UKiS with the HCT116 clone #1–10 (Supplementary Figs. 9 and 10). The resultant three sets of mutants showed significant increases in *TP53* mRNA levels, but the fold changes were equivalent among the three mutants, ~2-fold as compared with expression in the cells with the original human intron, which was much lower than that in the mutant with all 17 *Alu* elements deleted (Fig. 6c). To exclude the possibility that the ~2-fold *TP53* induction was simply due to intron shortening by removing the

1.4- to 1.9 kb *Alu* sequences in the three mutants, we also made another mutant clone from clone #1–10, in which ~2.5 kb of non-*Alu* sequence was removed from the intron ("~2.5 kb deletion of non-*Alu* sequences"). This mutant was associated with no significant change in the *TP53* mRNA level (Fig. 6c). Taken together, the suppressive effect on *TP53* expression seems to be caused not by a particular *Alu* element but by the accumulated effects from all or most of the 17 *Alu* elements. UKiS mutagenesis thus uncovered the unprecedented role of intronic retroelements in the context of native chromatin.

**Application of UKiS**. To test its general usability, we first applied UKiS to the human induced pluripotent stem (iPS) cell line 201B7 (Fig. 7a). We removed the first intron of *TP53* by the same protocol and materials that we used with HCT116 cells above (Figs. 2 and 3c) except that an additional gRNA was used in the first step of UKiS that cuts the site close to the left arm region to increase the substitution efficiency[24] (Supplementary Fig. 11). In the second step of UKiS, we still used only one gRNA corresponding to the right arm region because paired-gRNA cleavage could induce non-homologous end joining (NHEJ)-mediated deletion of the UKiS donor from both alleles, resulting in false positives during FACS sorting. Partially because of the single gRNA cleavage site, genotyping PCR after the second step indicated a lower probability of correctly targeted cell clones (three clones were correctly targeted, out of the total 10 cell clones obtained by FACS) than in HCT116 cells (Figs. 3c and 7a). The obtained three clones were then subjected to the heterologous SNP analysis to verify biallelic substitution (Supplementary Fig. 12). The iPS cell line 201B7 has a heterologous SNP site (A or G) at rs1641548 in the ninth intron of *TP53*. Genomic PCR and direct sequencing of the PCR products clarified that biallelic substitution had occurred at both steps of UKiS in the iPS cells and that the *TP53* intron deletion was achieved at both alleles in the three iPS cell clones. In these clones, the expression of two genes that are markers of pluripotency[25], *OCT4* and *NANOG*, was maintained (Fig. 7b), showing the dispensability of this intron for maintaining stemness of iPS cells. These data suggest that UKiS is a feasible platform for biallelic and large-scale genomic substitution in iPS cells.

Finally, we attempted to remove not one intron but all introns from three different non-essential human genes that are longer than *TP53* in HCT116 cells: CD44 molecule (Indian blood group) (*CD44*), MET proto-oncogene, receptor tyrosine kinase (*MET*), and amyloid beta precursor protein (*APP*) are 94, 126, and 290 kb in length and have 17, 21, and 18 exons, respectively (Fig. 7c–e). Here we also used two gRNAs for each gene locus in the first step of UKiS. Both steps of UKiS resulted in the isolation of the desired clones with high efficiency, leading to implementation of the complete removal of all the introns from the endogenous gene loci (Fig. 7c–e and Supplementary Figs. 13–15). Heterologous SNP sequencing analysis confirmed that the modified sequences were as expected and that the three cell clones obtained in the second step of UKiS in each case had achieved complete intron removal at both alleles (Supplementary Figs. 16–18). We then used these clones to examine the dispensability of introns for the transcription of these genes by RT-PCR (Fig. 7f–h). Interestingly, the three genes were affected differently. Intron-free *CD44* maintained its level of transcription almost equivalent to that in the original HCT116 cells, whereas complete intron removal brought *MET* transcription down to zero. The downsizing of *APP* by intron removal from 290 to 3.6 kb disrupted transcription drastically, but its transcription was still detectable and was ~20% of the level seen in the parental HCT116 cells. These observations were possible only with biallelic and gene-wide genomic substitution, which establishes the usability and advantage of UKiS.

## Discussion

This study demonstrates that UKiS is valuable for implementing systematic mutagenesis at a resolution of a single base pair within the entire region of endogenous human gene loci in living cells. We here performed the precise deletion of one or all introns from human genes, non-humanization of a human *TP53* intron into the mouse and zebrafish syntenic sequences, and specific removal of only particular retroelements from an intron, all of which were done in a biallelic manner. The removal of specific *Alu* elements and of all introns from the sub-megabase *APP* locus shows the technological advantages of UKiS. To our knowledge, precise intron removal has been systematically performed only for yeast genes[26–28]. Removal or substitution of introns in mouse genes has been performed with leaving behind loxP or a drug marker at the target sites[29], which resulted in possible artifacts in experiments to determine intron functionality. UKiS has allowed us to make a straightforward and unquestionable evaluation of the effects of these mutations, which thus provides insights into gene regulation by introns and retroelements. Notably, the UKiS scheme is applicable for human iPS cells, confirming its potential for various applications related to stem cell and regenerative medicine.

HR-mediated biallelic substitution has previously been performed in human iPS cells with different approaches. To our understanding, Byrne et al. first demonstrated that an endogenous gene on the scale of 10 kb could be substituted in a scarless and biallelic manner, with an efficiency of <1% even with the aid of single gRNA cleavage[30]. In their approach, only human genes that encode surface markers could be the target for substitution with the homologous mouse genes, as cells that had undergone successful biallelic substitution were collected by FACS with antibodies that bind specifically to the human or mouse markers. The development of a process that includes sequential positive and negative selection was the solution that UKiS provides to enable the substitution of any gene of interest and to obtain correctly targeted clones efficiently.

Ikeda et al. thus used a donor plasmid in the first (and positive) selection step of HR that contained CD19 variants and mCherry in a bicistronic form[31]. After enrichment for CD19-positive cells using antibodies against CD19 immobilized on magnetic beads, the resultant cell clones with brighter mCherry signals were chosen, as these were expected to have biallelic integration of the donor plasmid. In the second (and negative) selection step, the mCherry-expressing cells were transfected with the mutating payload, and then the resulting cells that were no longer bound to CD19 beads were collected as flow-through, which included cells in which the donor plasmid locus was successfully replaced with the desired sequence at both alleles. This method enabled substitutions as small as 1 bp and the integration of an ~1 kb reporter at both alleles of the target. We speculate, however, that this approach would not be able to accomplish gene-wide biallelic substitution, in contrast to UKiS, given our control experiment showing that the efficiency of the first step of UKiS was substantially lower when only the puromycin version of the UKiS donor was used in HCT116 cells (Supplementary Fig. 2). To increase the efficiency of the process and to ensure biallelic substitution in iPS cells, we incorporated simultaneous selection using two drugs (puromycin and blasticidin) with two different donors in the first step of UKiS. We thus combined the two-step HR-mediated positive and negative selection with usage of dual drug selection[32,33], which expectedly gave rise at high efficiency to cell clones with the expected biallelic substitution of the ~10 kb intron in iPS cells and sub-megabase genes in HCT116 cells. One drawback of the two-step HR strategy is the inability to target essential gene loci because their replacement in the first step with any selection marker donors would lead to cell death, although this limitation could be overcome by introducing cDNA

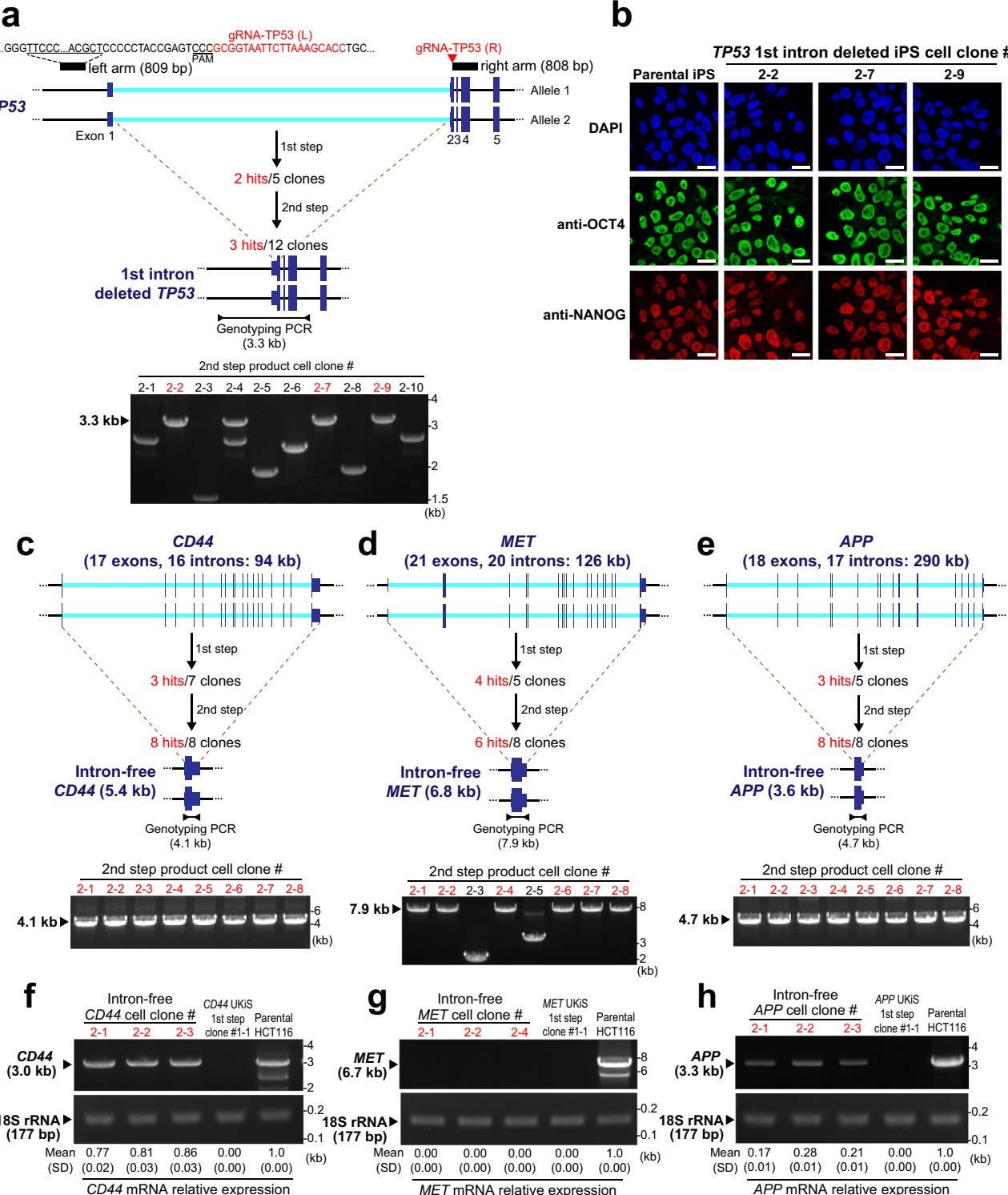

expression from episomes and/or viral vectors, which might allow the universal use of our knock-in system.

The sub-megabase-scale genome substitution in this study uncovered uncharacterized aspects of and also raised questions about the significance of endogenous introns and retroelements. It has been well acknowledged that the incorporation of introns into genes can change their expression, which has been used to improve expression from transgene cassettes[34]. Precise removal of every intron from all 280 yeast genes that contain at least one intron demonstrated that ~50 and 25% of the introns down- and

upregulate transcription, respectively[26]. One could thus speculate that intron removal can increase or decrease the transcription of human genes, although the empirical demonstration of this effect on endogenous genes in the context of native chromatin had yet to be performed at this scale until this study. It is also important to note that these results demonstrate that we have no means by which to predict the impact of the removal of any intron on the expression of its associated gene. Such predictions may be made based on accumulated ChIP-seq-related data on binding and marking by transcriptional and epigenetic factors, although some of these

**Fig. 7 UKiS application for iPS cells and longer noncoding targets. a** UKiS for deletion of the first intron of *TP53* in iPS cells. In addition to gRNA-TP53(R), the gRNA-TP53(L) was used, target sequence of which is highlighted in red, and the PAM sequence part is underlined. Intron deletion in the second step of UKiS was confirmed by junction genotyping PCR targeting the 3.3 kb region represented by a horizontal line flanked by two arrowheads. Clones having undergone successful intron removal are shown in red. The iPS cell clone used for the second step of UKiS (#1–2) was obtained by the first step of UKiS as shown in Supplementary Fig. 11. **b** Immunocytochemical staining for expression of pluripotent markers OCT4 (green) and NANOG (red) in the parental iPS cells and in the three mutant clones in which the first intron of *TP53* was deleted. Nuclear localization was confirmed by staining with DAPI (blue). Scale bar, 50 μm. Deletion of all introns from three human genes (**c**: *CD44*, **d**: *MET*, **e**: *APP*) using UKiS in HCT116 cells. Deletion of all introns was confirmed by junction genotyping PCR targeting the cDNA sequence from the first exon to the last exon of each gene, as represented by horizontal lines flanked by arrowheads, with the expected length of PCR genotyping amplicons indicated. Clones having successful recombination are shown in red. Representative image of the agarose gel electrophoresis for RT-PCR products targeting the (**f**) *CD44*, (**g**) *MET*, and (**h**) *APP* mRNA from original HCT116 cells, cell clones isolated after the first step of UKiS (Supplementary Figs. 13–15), and three clones corresponding to cells with intron-free genes of interest. The relative fold-changes of gene expression were calculated by quantifying band intensities using ImageJ[43] software. Values of mean ± SD of three independent RT-PCR experiments are shown. Source data are provided as a Source Data file.

biochemical events may work redundantly and/or be functionally irrelevant. Moreover, many human introns are heavily loaded with *Alu* elements, which frequently prevent unique mapping of ChIP-seq reads, especially with respect to *Alu* clusters, due to the presence of high sequence similarity among the ~1 × 10⁶ copies of *Alu* elements in the human genome[1].

Our ability to remove specific *Alu* elements from the *TP53* intron with UKiS unveiled a functional aspect of intronic *Alu* element clusters, transcriptional suppression. Whereas *Alu* elements that act as gene *cis*-regulatory elements are mostly located upstream of genes[35,36], it was recently reported that pairing of two *Alu* elements in introns can produce circular RNAs, which compete with the canonical splicing and generation of authentic mRNAs[37]. This could be the mechanism behind our observation, although the *Alu* cluster might possibly work as an intronic *cis*-element through transcription factor binding and/or epigenetic mechanisms that are known to underlie upstream *Alu*-mediated gene regulation[38]. Further systematic biallelic mutagenesis of the endogenous *TP53* intron by UKiS and omics analyses of the resultant mutant cell lines may unearth previously undescribed mechanistic features of *Alu*-mediated gene regulation. However, beyond *Alu* biology, UKiS can also help to uncover other functions of introns and also to formulate the principles for synthetic human genomics on how to design and re-build human genes in the context of native chromatin. As the removal of all introns from *CD44* and *APP* did not completely block mRNA expression of these genes in this study, the principles related to intron design will likely guide us toward a drastic downsizing of the human gene loci and, eventually, of the human chromosome without losing the basic cellular homeostasis[39].

Although this study focused on introns, UKiS is certainly applicable for general large-scale modification of any segments in the diploid genomes of humans and other higher organisms, which will no doubt allow us to address many questions about in vivo functional relationships among genome architecture, gene regulation, and cellular phenotypes. This scarless and site-specific method will also allow for sequential modification of different genomic regions toward the stepwise multi-functionalization of cells. UKiS may represent a pioneering platform for mammalian cell engineering for bio-production, disease modeling, therapeutics, and many other synthetic biology applications.

## Methods

**Analysis of intron length**. The lengths of human introns were obtained from NCBI RefSeq[40] annotation of assembly hg38 at the UCSC Genome Browser (https://genome.ucsc.edu/cgi-bin/hgTables)[41]. The length of each intron was derived from the exonStarts and exonEnds terms in the file, and the first intron was identified and extracted according to the value of the strand term. The rank of the length of the *TP53* first intron among all introns was calculated by sorting the length of each intron and deriving the position of the *TP53* first intron length

(10,754 bp) therein. Boxplots (Supplementary Fig. 1) were generated for all intron lengths and for all first intron lengths.

**Cell culture**. HCT116 (American Type Culture Collection, CCL-247) cells were maintained in McCoy's 5A Medium (Life Technologies) supplemented with 10% bovine growth serum (Biowest, S1820). 201B7 human iPS (RIKEN BRC, HPS0063) cells were maintained in StemFit AK02N medium (ReproCELL) in dishes coated with iMatrix-511 (Nippi). Both cell lines were cultured in a humidified incubator with 5% $CO_2$ and 95% air at 37 °C.

**Genomic DNA extraction**. Human genomic DNAs were extracted from HCT116, iPS cells and their derivative clones using the Monarch Genomic DNA Purification Kit (New England Biolabs) according to the manufacturer's instruction. The mouse genomic DNA was extracted and purified from the whole brain from C57BL/6N mouse by using DNeasy Blood & Tissue Kit (Qiagen) according to the manufacturer's instruction.

**Genomic PCR**. The target gene sequence was amplified with the extracted genomic DNA or the zebrafish genomic DNA (a gift from Dr. Katsumi Yamaguchi) by using KOD One PCR Master Mix (Toyobo) according to the manufacturer's instruction. All primers used for junction genotyping PCR of selected cell clones and for SNP typing are listed in Supplementary Tables 1 and 2, respectively. All SNPs referred in this article are registered in dbSNP151[42].

**RNA extraction and RT-PCR**. Total RNA was extracted using the RNeasy Mini Kit (Qiagen) and reverse-transcribed with the SuperScript IV First-Strand Synthesis System (Invitrogen) using oligo-dT and random hexamer as primers according to the manufacturers' instructions. The target gene sequence was amplified by using KOD One PCR Master Mix (Toyobo). All primers used for PCR are listed in Supplementary Table 3.

**Agarose gel electrophoresis**. PCR and RT-PCR products were subjected to 1% or 2% agarose gel electrophoresis by using Agarose S (Nippon Gene) in 1 × TAE (40 mM Tris-HCl, 20 mM acetic acid and 1 mM EDTA, pH 8.0). When required, the relative intensity of each band was measured using ImageJ[43] software (National Institutes of Health, Bethesda, MD). Uncropped and unprocessed scans of the gel images presented in the figures are supplied in the Source Data file.

**Sequencing**. DNA sequencing was performed by using the Big Dye Terminator v3.1 Cycle Sequencing Kit (Thermo Fisher) and an ABI 3100 DNA sequencer according to the manufacturers' instructions.

**Plasmids**. We constructed gRNA expression vectors by inserting annealed oligonucleotides including target sequences and adaptor sequences into *Bbs*I-digested pX330 (Addgene plasmid #42230).

For the construction of the first UKiS donor plasmid, we replaced the coding region for red fluorescent protein in the plasmid HR110PA-1 (Systems Biosciences) with that of GFP. To construct the second UKiS donor plasmid, we then took the resulting plasmid and replaced the sequence that encodes puromycin *N*-acetyltransferase with that which encodes blasticidin *S*-acetyltransferase. In addition, the left and right homology arms for the *TP53* locus, which were amplified by PCR using genomic DNA extracted from HCT116 cells, were inserted into the *Eco*RI and *Bam*HI sites, respectively, of the modified HR110PA-1 plasmids by using NEBuilder HiFi DNA Assembly (New England Biolabs).

For the construction of the mutating payloads for the full-length human intron, the intron deletion, the mouse intron, and the zebrafish intron, the PCR products were obtained by PCR amplification of genomic DNA extracted from HCT116 cells, cDNA prepared with total RNA from HCT116 cells, genomic DNA from mouse brain, and

that from the whole body of zebrafish as a template, respectively. Both homology arms used in the UKiS donor plasmids were also used. The mutating payloads and homology arms were assembled and integrated between the *Eco*RI and *Bam*HI sites on the original HR110PA-1 plasmid by using NEBuilder HiFi DNA Assembly.

For the construction of the mutating payload corresponding to retroelement-free and *Alu*-free *TP53* first introns, non-retroelement and non-*Alu* sequences from the *TP53* first intron were separately amplified from the human *TP53* first intron payload. Fragments of non-retroelement or non-*Alu* sequences were assembled by overlap-PCR. The resulting product and homology arms were integrated between the *Eco*RI and *Bam*HI sites on the original HR110PA-1 plasmid by using NEBuilder HiFi DNA Assembly.

For the construction of the mutating payloads corresponding to partial deletion of *Alu* sequences from the first intron of *TP53*, different regions of the *Alu*-deleted sequence and human *TP53* first intron sequence were amplified from the *Alu*-free mutating payload and human *TP53* first intron payload, respectively. Obtained products were assembled and integrated between the *Eco*RI and *Bam*HI sites on the original HR110PA-1 plasmid by using NEBuilder HiFi DNA Assembly.

For the construction of the mutating payload with deletion of ~2.5 kb of non-*Alu* sequences from the *TP53* first intron, four regions of the *TP53* first intron were amplified from the human *TP53* first intron payload that excluded non-*Alu* sequences. Obtained products were assembled and integrated between the *Eco*RI and *Bam*HI sites on the original HR110PA-1 plasmid by using NEBuilder HiFi DNA Assembly.

For the construction of the mutating payload for intron-free *APP*, *CD44*, and *MET*, the sequence from the left homology arm to the first exon and the sequence from the first exon to the last exon were amplified from genomic DNA and cDNA from HCT116, respectively, and integrated into the *Eco*RI and *Bam*HI sites on the original HR110PA-1 plasmid by using NEBuilder HiFi DNA Assembly. The amplified cDNA sequences of *APP*, *CD44*, and *MET* correspond to NM_000484.4, NM_001001389.2, and NM_000245.4, respectively.

All primers and oligonucleotides used for plasmid construction and sequencing of the constructed plasmids are listed in Supplementary Table 4.

**Transfection, antibiotic selection, and FACS.** Transfection into HCT116 cells and iPS cells was performed using FuGENE HD (Promega) and Lipofectamine Stem Transfection Reagent (Thermo Fisher), respectively, following the manufacturers' procedures. For the first step of UKiS, $2 \times 10^5$ HCT116 cells or $5 \times 10^5$ iPS cells were cultured in six-well plates. Approximately 14 h later, cells were transfected with the target gene-specific gRNA expression plasmid and the UKiS donor plasmids for the first step of UKiS. After incubation at 37 °C with 5% $CO_2$ for 12 h, the dual selection started with 1 μg/ml puromycin (Nacalai Tesque) and 10 μg/ml blasticidin (Funakoshi) for HCT116 cells or 0.5 μg/ml puromycin and 6 μg/ml blasticidin for iPS cells. The cell colonies emerging after 8 days were individually harvested with clean 200 μl pipette tips for single cloning and subsequent expansion. For the single donor plasmid experiment (Supplementary Fig. 2), all the procedures were the same except that the selection was performed with 1 μg/ml puromycin (Nacalai Tesque).

For the second step, $2 \times 10^5$ HCT116 cells or $5 \times 10^5$ iPS cells from individual clones that had UKiS donor sequences in both alleles were cultured in six-well plates. Approximately 14 h later, cells were transfected with the TL-gRNA expression plasmid and the payload plasmid of interest. Ten days after transfection, cells were trypsinized and filtered through a 40 μm cell strainer (Falcon) to obtain a single-cell suspension (~$10^6$ cells/ml). The suspended cells were sorted by Cell Sorter SH800 (Sony) or BD FACSMelody cell sorter (Becton, Dickinson and company) using SH800 Cell Sorter software (Sony) or BD FACSChorus software (Becton, Dickinson and company), respectively. The standard 488 nm lasers of these instruments were used to excite the GFP molecules, and green fluorescence was detected using 490 LP and 510/20 filters (Cell Sorter SH800) or 507 LP and 527/32 filters (BD FACSMelody Cell Sorter). The gating strategies used for cell sorting are shown in Supplementary Figs. 19–21.

**Off-target analyses.** The program COSMID[44] was used to identify any genomic loci that had sequences similar to the *TP53*-specific gRNA (gRNA-TP53(R)) and TL-gRNA target sequence. Off-target candidate loci were amplified by KOD One PCR Master Mix (Toyobo) according to the manufacturer's protocol. The obtained PCR products were purified and sequenced as described above. All primers used for off-target analysis are listed in Supplementary Table 5.

**Real-Time RT-PCR.** Total RNA was extracted using RNeasy Mini Kit (Qiagen) and reverse-transcribed by PrimeScript RT reagent Kit with gDNA Eraser (Takara). Real-time PCR was performed using TB Green Premix Ex Taq (Takara) and a Thermal Cycler Dice TP970 (Takara) and Thermal Cycler Dice Real Time System software (Takara) according to the manufacturer's protocols. Duplicates of reactions were run in three independent experiments. 18S rRNA was used as an internal control for normalization. The relative fold-changes were calculated by the ΔΔCt method. All primers used for RT-PCR are listed in Supplementary Table 6.

**Immunoblotting.** Protein samples were subjected to SDS-PAGE (10% polyacrylamide) and electroblotted onto an Immobilon-P membrane (Millipore). We run two SDS-PAGE gels for each experiment; one was for TP53 detection and the

other was for α-Tubulin. To both gels, the same amounts of the identical protein samples were loaded for normalization of TP53 expression levels using α-Tubulin as a control. Each blot was incubated overnight at 4 °C with an appropriate primary antibody as follows: anti-TP53 (1:2000, clone DO-1, sc-126, Santa Cruz Biotechnology) or anti-α-Tubulin (1:5000, clone 10G10, 017-25031, Wako). Next, each blot was then incubated with peroxidase-labeled anti-mouse IgG (1:5000, NA931V, GE Healthcare). Immunoreactive proteins were detected with Immobilon Western Chemiluminescent HRP substrate (Millipore) and the ImageQuant LAS 4000mini system (Cytiva) using Immunoblot image acquisition: Image Quant LAS 4000mini software (Cytiva). The band intensities were determined and expressed as pixel densities using ImageJ[43] software (National Institutes of Health, Bethesda, MD), and these values were normalized against the intensity of the corresponding control (α-Tubulin). Uncropped and unprocessed scans of the blot images presented in the figures are supplied in the Source Data file.

**Immunocytochemical staining and microscopy.** iPS cells were grown on 15 mm glass coverslips coated with iMatrix-511. Cells were fixed with 4% paraformaldehyde diluted in phosphate-buffered saline (PBS) for 15 min at room temperature, permeabilized with 0.1% Triton X-100 diluted in PBS and blocked with blocking buffer (0.1% Tween-20 and 1% bovine serum antigen diluted in PBS). Then, cells were stained with anti-OCT4 (ab19857, Abcam) and anti-NANOG (clone 1E6C4, sc-293121, Santa Cruz) diluted 500-fold in blocking buffer for 1 h at room temperature. The cells were then rinsed three times for 5 min each with 0.1% Tween-20 in PBS. Subsequently, the cells were stained with Alexa Fluor 488-conjugated goat anti-rabbit IgG (A-11008, Invitrogen) and Alexa Fluor 568-conjugated goat anti-mouse IgG (A-11004, Invitrogen) diluted 1000-fold in blocking buffer for 1 h at room temperature and then rinsed three times for 5 min each with 0.1% Tween-20 in PBS. Finally, cells were stained with 2 μg/μl DAPI (Dojindo, D523) in PBS for 5 min at room temperature. Images were acquired with an LSM780 confocal microscope (Zeiss) and Zen Software Black Edition (Zeiss).

**Statistics and reproducibility.** Regarding the sample size, we decided to collect 5–10 cell clones after the drug selection and FACS sorting at the 1st and 2nd steps of UKiS, respectively. For cell clone characterization, we decided that three different cell clones having the expected genome modification after the 2nd step of UKiS were examined as biological triplicates for every mutant. Since the 3 clones of every mutant were randomly chosen, no bias should be present to evaluate the effects of genome modification that we introduced. Experiments were repeated three times for confirmation and characterization of obtained cell clones (the junction PCR/agarose gel electrophoresis and RT-PCR/real-time RT-PCR/ immunoblotting/immunocytochemical staining, respectively). UKiS to each target gene was performed in a single experiment, including transfection, dual selection, and FACS sorting. Off-target analyses and SNP typing were performed one time. No data was excluded. Most of the experiments were performed by different co-authors. The data were also processed and re-evaluated by more than two authors. Statistical significance was determined by using a two-tailed Student's t-test; $p < 0.05$ was considered to be significant.

**Reporting summary.** Further information on research design is available in the Nature Research Reporting Summary linked to this article.

## Data availability
The data supporting the findings of this study are available within the main body of this manuscript and its Supplementary Information files. The source data underlying Figs. 2c, 3b–e, 5a, c–e, 6b, c, and 7a, c–h, as well as Supplementary Figs. 2b, 3c, 4c, 6, 8, 9, 11b, 13b, 14b, and 15b are provided as a Source Data file. Source data are provided with this paper.

## Code availability
The script used for the intron-length analysis (Supplementary Fig. 1) has been deposited to GitHub: https://github.com/Taichi-Akase/UKiS.

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

## Acknowledgements

We thank Drs. Jef D. Boeke, Fumiaki Hamazato, Takahide Yokoi and the members of the Aizawa Laboratory for their fruitful discussions and critical reading and Dr. Katsumi Yamaguchi for providing the zebrafish genomic DNA. We also thank the Open Facility Center, the Cell Biology Center (Institute of Innovative Research) and the Open Research Facilities for Life Science and Technology, Tokyo Institute of Technology for providing the service and sharing their instruments. This work was supported by JST CREST "Large-scale genome synthesis and cell programming" (Grant Number: JPMJCR18S5) to Y.A. A grant-in-aid (IBUNNYAYUGO) was provided to Y.A. by the Kanagawa Prefectural Government for Integration of Advanced Multidisciplinary Research Activities.

## Author contributions

T.O., T.T., and Y.A. formulated the concept of this research and designed the experiments. T.O., T.T., T.A., S.K., and H.K. performed the experiments. S.K. and T.A. provided the computational analyses. T.O. and Y.A. wrote the manuscript. All authors discussed and commented on the final draft of the manuscript.

## Competing interests

Patent application has been filed for the technology described in this manuscript: patent applicant: - Logomix, Inc. - Tokyo Institute of Technology name of inventor(s): - T.O. and Y.A. are named as the inventors of the patent for technologies related to UKiS. application number: - JP2020-068266 - WO2021/206054 A1 status of application: - In process of entering into the National Phase specific aspect of manuscript covered in patent application - The methodology of UKiS Y.A. is a co-founder and the chief scientific officer of Logomix, Inc. and T.O. is an employee of Logomix, Inc. The remaining authors declare no competing interests.
