## [Peer Review File · Nature Communications]

Reviewers' Comments:

Reviewer #1:

Remarks to the Author:

In the manuscript NCOMMS-20-42781-T by Ohno et al., the authors describe a tag-and-exchange strategy specifically applied to the replacement of intron 1 of the human p53 gene in HCT116 cells. To enhance the exchange step the method employs CRISPR gRNAs which the authors demonstrate to have low off-target activity. They describe exchange with syntenic introns from non-human species, as well as engineered human repeat-free introns. These exchanges result in differential expression of p53 and p53-related genes, leading the authors to conclude that the deleted repeat sequences are involved in p53 regulation. These applications are noteworthy, as the exchanges constructs are large and highly modified, which would prove difficult to achieve through multiple rounds of gene editing, especially considering the repetitive nature of most intron sequences. The most notable data is therefore shown in Figure 5.

While the application of the method is certainly interesting, it is not new or novel as presented in the manuscript. The targeting strategy used, called "double replacement" or "tag and exchange", was first published nearly 30 years ago by multiple groups (Askew, 1993; Stacey 1994; Wu, 1994; reviewed in PMID: 9809673). The intermediate cell lines have been shown to be valuable in the generation of allelic series (PMID: 8391633, 9636277), or species-specific gene replacement such as the 2.972 kb human alpha-lactalbumin in mouse ES cells (PMID: 8289781). Moreover, the application of tag and exchange to modify intronic regions was already demonstrated for the Col1A1 gene in 1998 (PMID: 9584177).

The method outlined in Figure 1 makes use of a unique nuclease cleavage site deposited by the UKiS donor in the first round of targeting. Decades earlier, this same strategy to stimulate a double-strand break and improve the selection marker exchange originally employed I-SceI target site deposition in the first targeting step (PMID: 9488460). As cited by Ohno et al., the strategy has recently been adopted to CRISPR (Ikeda et al., 2018). This approach was wisely selected - as the authors note - since it standardizes gRNAs to reduce variation in off-target effects between clones edited at different loci, but is by no means novel. Finally, previous examples make use of two-marker or even two-color selection (PMID: 16258059) to enrich for biallelic targeting events.

The authors state the method as being "universal", but do not present data for efficient tag and exchange in other cell types or at other loci to support the claim. Moreover, essential genes cannot be bi-allelically targeted using standard selection marker knock-in strategies.

In support of the method, it should be made clear that it is not only specific to intron editing, despite the title and descriptions throughout the main text. However, given the repetitive nature of non-coding DNA, it is uniquely adapted to it. The method is in fact applicable to tag and exchange any genomic region which is flanked by unique sequence suitable for nuclease cleavage and homologous recombination. Rather than presenting the method as something entirely novel, it is recommended that the authors revise the text to focus on their new improvements and applications (species-specific and repeat element-free introns).

Technically, the manuscript would be supported by genotyping beyond simple PCR, such as sequence (not just junctions) or structure (ie. Southern blot) of the edited alleles, and copy number analysis for biallelic editing. As a method, more detail on the vectors and procedure is required for reproduction.

Reviewer #2:

Remarks to the Author:

The authors are interested in intron function and have developed a dual drug selection method to replace whole introns in both alleles of a gene. They use their method, UKiS, to remove the first intron of human TP53 and also to replace it with the first intron of mouse Tp53 and zebrafish tp53. Intron deletion and the mouse intron increased TP53 mRNA levels. They also designed an intron with no retrotransposons and find a large increase in TP53 transcription, which was also seen in

the construct in which they deleted only the Alu elements, and the levels of transcript seen in genes regulated by TP53 also increased.

This is a clever technique and I recommend publication of this manuscript.

A few comments.

1. Is it possible that the increase in transcription seen with the retroelement removed introns is only because the intron is shorter? The Alu-free intron is 5.5kb; if any 5.5kb sequence were used would transcription levels go up?
2. It should be noted that the levels of downstream genes do not depend directly on the Alu elements in the intron but rather on the level of TP53.

minor:

1. A few places have awkward grammar (e.g. "one of the clones was commonly used", "mutation types are difficult to be created by")
2. Figure 4C "Humain" should be "Human"

Point-by-point responses to comments from the editor and the reviewers

(our replies are in a blue font)

The main concerns from the Editor's letter

> *You will see that, while the reviewers find your work of interest, they raise substantive concerns that cast doubt on the advance your findings represent over similar genome editing strategies in use over the past decades and the strength of the novel conclusions that can be drawn at this stage.*

Our reply:

One of the main reasons for your decision was the point underlined above. As this point was brought up only by Reviewer #1, we have addressed this comment below, but we also would like to note here that the references Reviewer #1 cited with respect to this point are all about genome engineering in mouse cells, not in human cells, except for the ones that we cited in the initially submitted manuscript. As you know, genome engineering in mouse embryonic stem (ES) cells is much more efficient than it is in human cells. Statements in the literature to this effect include the following:

- "High rates of homologous recombination between an exogenous donor vector and the endogenous genome are routinely achieved in mouse ES cells..." (Genome Biol 16(1):109, 2015)
- "The discovery of mouse embryonic stem (ES) cells >20 years ago represented a major advance in biology and experimental medicine, as it enabled the routine manipulation of the mouse genome." (Genes Dev 19:1129–1155, 2005)
- "Significant differences between mouse and human ES cells have hampered the development of homologous recombination in human ES cells." (Nat Biotechnol 21:391–321, 2003)

In this sense, we suggest that the studies cited by Reviewer #1 in fact support the novelty of our study with respect to the manipulation of the human genome. Our method (UKiS; Universal Knock-in System) enables precise and biallelic genome modification of >10-kb regions at single-nucleotide resolution in human cells, which is much improved from the previous work on human genome engineering and which we have now emphasized in the Discussion. Furthermore, we have added new data showing its applicability to human induced pluripotent stem (iPS) cells (Figure 7a and 7b, Supplementary Figures 11 and 12), further confirming the applicability of UKiS in human cells.

The other concern mentioned in your letter was the novelty of our biological findings (dotted-underlined in your comment; please see above), specifically, we believe, regarding the roles of *Alu* transposons in *TP53* expression. During the revision process, we performed

additional mutagenesis experiments (Figure 6c, Supplementary Figures 9 and 10), which led us to a novel implication of *Alu* functionality that the suppressive effect of the intronic *Alu* elements on *TP53* expression might be caused not by a particular *Alu* element but by the accumulated effects from all 17 *Alu* elements. We believe this finding underscores the potential role that UKiS could play in characterizing unprecedented roles of introns and retroelement.

From Reviewer #1 (Remarks to the Author):

Comment #1-1:

In the manuscript NCOMMS-20-42781-T by Ohno et al., the authors describe a tag-and-exchange strategy specifically applied to the replacement of intron 1 of the human p53 gene in HCT116 cells. To enhance the exchange step the method employs CRISPR gRNAs which the authors demonstrate to have low off-target activity. They describe exchange with syntenic introns from non-human species, as well as engineered human repeat-free introns. These exchanges result in differential expression of p53 and p53-related genes, leading the authors to conclude that the deleted repeat sequences are involved in p53 regulation. These applications are noteworthy, as the exchanges constructs are large and highly modified, which would prove difficult to achieve through multiple rounds of gene editing, especially considering the repetitive nature of most intron sequences. The most notable data is therefore shown in Figure 5.

Our reply:

We are grateful for the reviewer's careful reading and appreciation of our approach. To further explore the data originally shown in Figure 5, the revised manuscript now has new data from more detailed deletions of *Alu* elements (Figure 6c, Supplementary Figure 9), leading to new implication that "the suppressive effect on *TP53* expression seems to be caused not by a particular *Alu* element but by the accumulated effects from all or most of the 17 *Alu* elements" (Lines 236–238).

Comment #1-2:

While the application of the method is certainly interesting, it is not new or novel as presented in the manuscript. The targeting strategy used, called "double replacement" or "tag and exchange", was first published nearly 30 years ago by multiple groups (Askew, 1993; Stacey 1994; Wu, 1994; reviewed in PMID: 9809673). The intermediate cell lines have been shown to be valuable in the generation of allelic series (PMID: 8391633, 9636277), or species-specific gene replacement such as the 2.972 kb human alpha-lactalbumin in mouse ES cells (PMID: 8289781). Moreover, the application of tag and exchange to modify intronic regions was already demonstrated for the Col1A1 gene in 1998

(PMID: 9584177).

The method outlined in Figure 1 makes use of a unique nuclease cleavage site deposited by the UKiS donor in the first round of targeting. Decades earlier, this same strategy to stimulate a double-strand break and improve the selection marker exchange originally employed I-SceI target site deposition in the first targeting step (PMID: 9488460). As cited by Ohno et al., the strategy has recently been adopted to CRISPR (Ikeda et al., 2018). This approach was wisely selected - as the authors note - since it standardizes gRNAs to reduce variation in off-target effects between clones edited at different loci, but is by no means novel. Finally, previous examples make use of two-marker or even two-color selection (PMID: 16258059) to enrich for biallelic targeting events.

Our reply:

We appreciate the context provided by the reviewer with respect to previous genome engineering technologies. We note, however, that the studies that this reviewer has cited all refer to genome engineering in mouse cells, not human cells. Indeed, genome manipulation in mouse ES cells or embryos is more feasible than in human cells, in general (please see our related citations above).

To accomplish this large-scale genome modification in human cells, we combined the three different strategies that this reviewer mentioned (double replacement, two-marker selection and CRISPR-mediated homologous recombination enhancement), improving biallelic substitution of the ~10-kb region in human HCT116 cells in the initially submitted manuscript. To the best of our knowledge, this is the first report that combined these three strategies for human genome engineering. We have added explanations about this improvement in the Discussion by citing now two papers on biallelic and scarless genome modification in human cells (Lines 294–322). In addition, we have now added new pieces of evidence on biallelic and precise substitution of this 10-kb region in human iPS cells (Figure 7a and 7b). Furthermore, we have also added data showing biallelic substitution of up to 290-kb regions in three additional genes in HCT116 cells (Figure 7c-h). This revised manuscript hopefully more clearly demonstrates the usefulness of combining these three pre-existing technologies for large-scale and biallelic engineering in human cells.

Comment #1-3:

The authors state the method as being “universal”, but do not present data for efficient tag and exchange in other cell types or at other loci to support the claim. Moreover, essential genes cannot be bi-allelically targeted using standard selection marker knock-in strategies.

Our reply:

Again, we appreciate the insightful comments from this reviewer. We, too, were initially concerned that the term “universal” might be an overstatement. The addition of our new data, however, supports to the use of this term. We now have shown that this method can be used to target 10-kb and 290-kb genomic regions at different gene loci for efficient substitution in human iPS and HCT116 cells, respectively (Figure 7, Supplementary Figures 11 and 13-15). We agree that although essential genes cannot be target directly by our method, the incorporation of a rescuing essential gene cassette at a different locus temporally may allow the biallelic and scarless modification of the original locus of the essential gene with our method. This drawback and its possible solution are now described in the Discussion (Line 322-326). Although we have yet to carry out such experiments, as we are currently applying this method at many other loci and in more complex ways, we hope to keep using the name “universal knock-in system” or “UKiS”.

Comment #1-4:

In support of the method, it should be made clear that it is not only specific to intron editing, despite the title and descriptions throughout the main text. However, given the repetitive nature of non-coding DNA, it is uniquely adapted to it. The method is in fact applicable to tag and exchange any genomic region which is flanked by unique sequence suitable for nuclease cleavage and homologous recombination. Rather than presenting the method as something entirely novel, it is recommended that the authors revise the text to focus on their new improvements and applications (species-specific and repeat element-free introns).

Our reply:

We appreciate that the reviewer has suggested a broader application potential for UKiS than what our original manuscript described. With this supportive suggestion and our newly added data in the Results, we have now emphasized that our method is appropriate for mutagenesis in introns and retroelements by revising the Title, Abstract, and Discussion.

Comment #1-5:

Technically, the manuscript would be supported by genotyping beyond simple PCR, such as sequence (not just junctions) or structure (ie. Southern blot) of the edited alleles, and copy number analysis for biallelic editing.

Our reply:

In response to this comment, we have included in the revised manuscript data from long PCR and subsequent direct sequencing that reaches from the replaced region beyond the homologous arm to heterozygous SNP sites that were used for SNP typing to check biallelic editing (Figure 4). The cell clones that were used for the downstream characterization were

all subjected to long PCR and SNP typing, which confirmed that these clones were all biallelically modified (Supplementary Figures 5, 7, 10, 12, and 16-18).

Comment #1-6:

As a method, more detail on the vectors and procedure is required for reproduction.

Our reply:

We apologize for the absence of details in our original description. The plasmid section and supplementary information in the revised manuscript have been amended to allow reproduction of the plasmid construction.

Reviewer #2 (Remarks to the Author):

Comment #2-1:

The authors are interested in intron function and have developed a dual drug selection method to replace whole introns in both alleles of a gene. They use their method, UKiS, to remove the first intron of human TP53 and also to replace it with the first intron of mouse Tp53 and zebrafish tp53. Intron deletion and the mouse intron increased TP53 mRNA levels. They also designed an intron with no retrotransposons and find a large increase in TP53 transcription, which was also seen in the construct in which they deleted only the Alu elements, and the levels of transcript seen in genes regulated by TP53 also increased. This is a clever technique and I recommend publication of this manuscript.

Our reply:

We are grateful for the reviewer's careful reading and appreciation of our manuscript.

A few comments.

Comment #2-2:

1. Is it possible that the increase in transcription seen with the retroelement removed introns is only because the intron is shorter? The Alu-free intron is 5.5kb; if any 5.5kb sequence were used would transcription levels go up?

Our reply:

This is a very important point, which we agree needs to be addressed. We have thus made a series of four additional *Alu* deletion mutants (Figure 6c, Supplementary Figure 9). The first three consist of deletions of a subset of the *Alu* sequences (group I *Alu* deletion, group II *Alu* deletion, and group III *Alu* deletion), each of which is missing four to six *Alu* elements and is shorter than the full-length intron by 1.8–1.9 kb. The other mutant consists of an ~2.5-kb deletion of non-*Alu* sequences. Comparisons among multiple clones of these four mutants clearly demonstrated that *Alu* deletion, and not shortening of the intron in general,

was responsible for the increase in *TP53* transcription. Also, the 1.8- to 1.9-kb deletions corresponding to a subset of *Alu* elements resulted in more modest suppression of *TP53* as compared with deletion of all *Alu* elements. These new data led us to an interesting implication that *TP53* suppression seems to be caused not by a particular *Alu* element but by the accumulated effects from all or most of the 17 *Alu* elements. We appreciate the reviewer for providing us with the opportunity to go one step further in understanding our initial observation.

Comment #2-3:

2. *It should be noted that the levels of downstream genes do not depend directly on the Alu elements in the intron but rather on the level of TP53.*

Our reply:

We apologize for the confusion that our initial wording resulted in. We have now indicated that “the intronic *Alu* sites have a substantial impact on signaling in the *TP53* pathway via tuning *TP53* expression” (Lines 221–222).

Comment #2-4:

1. A few places have awkward grammar (e.g. “one of the clones was commonly used”, “mutation types are difficult to be created by”)
2. Figure 4C “Humain” should be “Human”

Our reply:

We appreciate the care with which this reviewer has read the manuscript. Experienced professional scientific editors who are native English speakers have reviewed the entire manuscript to correct not only typographical errors but also as many grammatical and rhetorical errors as possible.

Reviewers' Comments:

Reviewer #1:

Remarks to the Author:

The authors have made a commendable effort to extend the data and revise the text from its original version. The inclusion of iPSC genome editing and targeting of additional genes in HTC116 cells supports a broader application of the method. Genotyping of heterozygous SNPs improves confidence of homozygous editing, and details on vector construction are provided. Overall, the quality of the paper is greatly improved.

On one minor point, the authors should remove the redundant statement "...that does not cause them to lose their stemness." (line 259-260). It is more relevant to demonstrate this after removing introns from TP53 (line 258) and not after gene editing.

Still, this Reviewer maintains that the application to study non-coding regions is the main point of interest of the paper. Despite the arguments of the authors regarding mouse/human applications, that alone does not instill novelty. All aspects (tag and exchange, dual selection to enrich biallelic editing, all using CRISPR) have been previously demonstrated in human cells in some combination or another.

This said, the overall language regarding novelty is tempered and the revised discussion better addresses similarities with past strategies; it is acceptable to inform readers who may be new to the gene editing field. In the end, the execution of the author's strategy is solid and may provide interesting insights into cell biology and gene expression. With the above minor editorial change, the revised manuscript is recommended for publication.

Reviewer #2:

Remarks to the Author:

The authors have addressed my concerns.

NCOMMS-20-42781B

Point-by-point responses to comments from the editor and the reviewers

(Our replies are in a blue font)

Reviewer #1 (Remarks to the Author):

The authors have made a commendable effort to extend the data and revise the text from its original version. The inclusion of iPSC genome editing and targeting of additional genes in HTC116 cells supports a broader application of the method. Genotyping of heterozygous SNPs improves confidence of homozygous editing, and details on vector construction are provided. Overall, the quality of the paper is greatly improved.

On one minor point, the authors should remove the redundant statement "...that does not cause them to lose their stemness." (line 259-260). It is more relevant to demonstrate this after removing introns from TP53 (line 258) and not after gene editing.

Our reply:

We appreciate the reviewer's careful reading. We completely agree and removed "that does not cause them to lose their stemness" in the final revision.

Still, this Reviewer maintains that the application to study non-coding regions is the main point of interest of the paper. Despite the arguments of the authors regarding mouse/human applications, that alone does not instill novelty. All aspects (tag and exchange, dual selection to enrich biallelic editing, all using CRISPR) have been previously demonstrated in human cells in some combination or another.

This said, the overall language regarding novelty is tempered and the revised discussion better addresses similarities with past strategies; it is acceptable to inform readers who may be new to the gene editing field. In the end, the execution of the author's strategy is solid and may provide interesting insights into cell biology and gene expression. With the above minor editorial change, the revised manuscript is recommended for publication.

Our reply:

All the comments and suggestions from this reviewer indeed gave us the great opportunity to improve the entire manuscript. We all authors show our greatest appreciation of her or his time, effort and support for our work.

Reviewer #2 (Remarks to the Author):

The authors have addressed my concerns.

Our reply:

Again, we can't show our appreciation enough to this reviewer as well. All the comments and suggestions from this reviewer indeed gave us the great opportunity to improve the entire manuscript. We all authors show our greatest appreciation of her or his time, effort and kindness for our work.